# A Novel Multi-Strain E3 Probiotic Formula Improved the Gastrointestinal Symptoms and Quality of Life in Chinese Psoriasis Patients

**DOI:** 10.3390/microorganisms12010208

**Published:** 2024-01-19

**Authors:** Pui Ling Kella Siu, Chi Tung Choy, Helen Hoi Yin Chan, Ross Ka Kit Leung, Un Kei Chan, Junwei Zhou, Chi Ho Wong, Yuk Wai Lee, Ho Wang Chan, Claudia Jun Yi Lo, Joseph Chi Ching Tsui, Steven King Fan Loo, Stephen Kwok Wing Tsui

**Affiliations:** 1Microbiome Research Centre, BioMed Laboratory Company Limited, Hong Kong, China; kellasiu@biomed.com.hk (P.L.K.S.);; 2Hong Kong Institute of Integrative Medicine, Faculty of Medicine, The Chinese University of Hong Kong, Hong Kong, China; 3Dermatology Centre, CUHK Medical Centre, The Chinese University of Hong Kong, Hong Kong, China; 4School of Biomedical Sciences, Faculty of Medicine, The Chinese University of Hong Kong, Hong Kong, China; 5Centre for Microbial Genomics and Proteomics, The Chinese University of Hong Kong, Hong Kong, China; 6Hong Kong Bioinformatics Centre, The Chinese University of Hong Kong, Hong Kong, China

**Keywords:** psoriasis, gastrointestinal problems, gut microbiome, probiotics, gut–skin axis

## Abstract

Psoriasis is a chronic immune-mediated inflammatory disease affecting the skin and other systems. Gastrointestinal disease was found to be correlated with psoriasis in previous studies and it can significantly affect the quality of life of psoriasis patients. Despite the importance of the gut microbiome in gut and skin health having already been demonstrated in many research studies, the potential effect of probiotics on GI comorbidities in psoriasis patients is unclear. To investigate the effects of probiotics on functional GI comorbidities including irritable bowel syndrome, functional constipation, and functional diarrhea in psoriasis patients, we conducted a targeted 16S rRNA sequencing and comprehensive bioinformatic analysis among southern Chinese patients to compare the gut microbiome profiles of 45 psoriasis patients over an 8-week course of novel oral probiotics. All the participants were stratified into responders and non-responders according to their improvement in GI comorbidities, which were based on their Bristol Stool Form Scale (BSFS) scores after intervention. The Dermatological Life Quality Index (DLQI) score revealed a significant improvement in quality of life within the responder group (DLQI: mean 10.4 at week 0 vs. mean 15.9 at week 8, *p* = 0.0366). The proportion of psoriasis patients without GI comorbidity manifestation at week 8 was significantly higher than that at week 0 (week 0: Normal 53.33%, Constipation/Diarrhea 46.67%; week 8: Normal 75.56%, Constipation/Diarrhea 24.44%, *p* = 0.0467). In addition, a significant difference in the gut microbiome composition between the responders and non-responders was observed according to alpha and beta diversities. Differential abundance analysis revealed that the psoriasis patients exhibited (1) an elevated relative abundance of *Lactobacillus acidophilus*, *Parabacteroides distasonis*, and *Ruminococcus bromii* and (2) a reduced relative abundance of *Oscillibacter*, *Bacteroides vulgatus*, *Escherichia* sp., and *Biophila wadsworthia* after the 8-week intervention. The responders also exhibited a higher relative abundance of *Fusicatenibacter saccharivorans* when compared to the non-responders. In summary, our study discovers the potential clinical improvement effects of the novel probiotic formula in improving GI comorbidities and quality of life in psoriasis patients. We also revealed the different gut microbiome composition as well as the gut microbial signatures in the patients who responded to probiotics. These findings could provide insight into the use of probiotics in the management of psoriasis symptoms.

## 1. Introduction

Psoriasis is a chronic immune-mediated inflammatory skin disease with an estimated 60 million patients worldwide [1,2], characterized by erythematous plaques covering the extensor surfaces, scalp, and lumbosacral region [1]. The southern Chinese population is less susceptible to psoriasis compared to other regions. The local prevalence of psoriasis in Hong Kong is 0.3–0.6% [3], lower than the global prevalence of 2–3% [4,5]. Although the primary manifestation of psoriasis is on the skin, it is now recognized that this disorder has systemic implications with complications such as cardiovascular disease, diabetes mellitus, obesity, non-alcoholic fatty liver disease (NAFLD), and inflammatory bowel disease (IBD) [6,7,8,9,10], which will significantly affect patients’ quality of life. Research studies have revealed a higher prevalence of gastrointestinal (GI) comorbidities in psoriasis patients than that in the general population, the risk of which will also increase along with the psoriasis severity [7]. It is still unclear whether psoriasis is a predisposing factor in or a consequence of a disturbed gastrointestinal environment, but the coexistence of psoriasis and GI comorbidities is well recognized by the scientific community [11].

Gut dysbiosis is commonly observed in psoriasis patients with GI comorbidities, including lower microbiome diversity, the overgrowth of harmful microorganisms, and a decrease in beneficial microorganisms [12]. Gut dysbiosis is also observed in other inflammatory skin diseases, such as atopic dermatitis (AD) and rosacea [13]. The association between gut dysbiosis and skin conditions was conceptualized and postulated as the “gut–skin axis” [13]. The gut microbiome will interact extensively with the skin to modulate local and systemic inflammation through the immune system [14,15]. In general, gut dysbiosis will increase the production of harmful metabolites, such as proinflammatory cytokines and toxins. These metabolites will reach the skin via the circulatory system, contributing to pathogen overgrowth, provoking skin inflammation, and ultimately exacerbating the progression of skin conditions like psoriasis [16]. Therefore, it is crucial to maintain homeostasis in the gut microenvironment, not only to maintain gut health but also to boost host immunity and foster optimal skin health.

The primary purposes of current psoriasis treatment are the alleviation of psoriasis symptoms, an improvement in quality of life, and the prevention of disease progression [17,18]. For mild psoriasis patients, topical corticosteroids are commonly used to reduce skin inflammation and inhibit skin cell proliferation [19]. However, this method is less effective in treating severe psoriasis patients. Therefore, the U.S. and European guidelines recommended the combination use of biologics, oral agents, and phototherapy methods for the treatment of severe psoriasis patients [20,21,22,23]. Nevertheless, previous studies have shown the gut microbiome composition of psoriasis patients receiving these treatments was significantly different from that of healthy populations [24]. An increased abundance of *Firmicutes* and a decreased *Bacteroidetes* abundance were observed in psoriasis patients receiving phototherapy and biological treatment [25]. Another study also observed decreased Faecalibacterium abundance in psoriasis patients receiving topical steroid treatment [26]. Therefore, the current management plan for psoriasis may not be able to restore the homeostasis of the gut microbiome. With the recognition of the essential role of the gut microbiome in psoriasis and psoriasis-related GI comorbidities, probiotics may be a potential therapeutic approach to restoring gut and skin health through maintaining gut symbiosis.

Our previous finding suggested that probiotics containing *Bifidobacterium* and *Lactobacillus* species can restore the gut microbiome balance and improve the Dermatological Life Quality Index (DLQI) and Psoriasis Area and Severity Index (PASI) in psoriasis without any adverse effects [27]. Although there has been growing evidence of studies focusing on the effects of probiotics in the management of psoriasis, the effects of probiotics on improving psoriasis-related GI comorbidities are still unclear. In this study, we aimed to investigate the potential clinical effects of probiotics in the management of GI comorbidities of psoriasis patients. Specifically, we sought to explore the diversity in the gut microbiome composition and identify distinctive gut microbiome signatures among psoriasis patients who exhibited favorable responses to probiotics. To achieve this, we retrospectively allocated them into responder and non-responder groups based on their improvement in GI symptoms after an 8-week probiotic intervention. In addition, we aimed to identify potential gut microbial signatures between these two groups using targeted 16S rRNA sequencing. These findings could help to evaluate, refine, and improve the clinical efficacy of probiotics as an adjuvant therapeutic option to alleviate psoriasis-related GI comorbidities to supplement the current psoriasis management plan.

## 2. Materials and Methods

### 2.1. Subject Recruitment amd Study Design

A total of 58 adults (18–65 years old) with psoriasis of Chinese ethnicity were recruited through the collaboration between the Hong Kong Psoriasis Patients Association, Hong Kong Society of Gut Microbiome, and BioMed Microbiome Research Centre. The inclusion and exclusion criteria followed our previous published study on psoriasis [27]. All participants came to screening visits at week 0 (baseline) and at week 8 ± 3 days for initial assessments. Bristol Stool Form Scale (BSFS), PASI, and DLQI (psoriasis) scores were assessed, and the gut microbiome was investigated at week 0 and week 8 after receiving oral multi-strain probiotics. All participants were allowed to continue their usual medication or topical maintenance therapy for psoriasis during the trial. Intervention compliance and adverse events were assessed at the visits. The participants were instructed to collect fecal specimens at baseline and week 8.

This study was conducted according to the guidelines of the Declaration of Helsinki and was approved by the Research Ethics Committee of the Hong Kong Doctors Union (protocol number HKSGM-2020AD-Study-protocol-vl-20220211).

### 2.2. Assessment Tools

#### 2.2.1. Bristol Stool Form Scale (BSFS)

The BSFS is a self-reported 7-point scale used to assess the quality of stool based on its shape and texture to distinguish watery, loose, and hard stool, which has been extensively validated and proved to be a reasonably reliable instrument to record the status of bowel movements [28]. The BSFS score ranges from Type 1 (hardest) to Type 7 (softest), in which Types 3–5 are considered normal, Types 1–2 constipation, and Types 6–7 diarrhea.

#### 2.2.2. Psoriasis Area and Severity Index (PASI)

The PASI includes an assessment of four body regions (head and neck, trunk, upper limbs, and lower limbs) by recording the percentage of affected skin in each region (area score), as well as the severity of psoriasis by measuring the intensity of redness, thickness, and scaling of these four regions [29]. The PASI score of each region ranges from 0 to 6, with the psoriasis severity increasing with a higher score [30]. All the patients involved in this study were evaluated by a board-certified dermatologist.

#### 2.2.3. Dermatology Life Quality Index (DLQI)

The DLQI is a simple, self-rated, user-friendly, and well-validated questionnaire consisting of 10 questions covering 6 sections, including patients’ perception of their symptoms and feelings; of their impact on their daily activities, leisure, work and school, personal relationships; and of treatment [29]. Each question is scored on a four-point Likert scale and the sum of the scores extrapolates to the effect of the skin disease on the patient’s life.

### 2.3. Library Preparation and 16S rRNA Sequencing

All the stool samples were homogenized in PurSafe^®^ DNA and RNA preservative (Puritan, Pittsfield, ME, USA) and were beaten with 425–600 µm glass beads (Sigma-Aldrich, Saint Louis, MO, USA) for 1 h according to the instructions from the manufacturer. A DNeasy Blood & Tissue Kit (QIAGEN, Hilden, Germany) was used to isolate the microbial DNA from the stool samples. The concentration of the extracted DNA from each sample was quantified using a Qubit™ dsDNA HS Assay Kit (Life Technologies, Carlsbad, CA, USA) and a Qubit 3 Fluorometer (Thermo Fisher Scientific, Waltham, MA, USA). The amplicon library included 515F (5′-GTGCCAGCMGCCGCGG-3′)/907R (5′-CCGTCAATTTCMTTTRAGTTT-3′) and a primer pair spanning and targeting the V4–V5 hypervariable region of the 16S rRNA genes, as well as the adapter sequences, multiplex identifier tags, and library keys. The 16S rRNA gene sequencing was performed using the Illumina MiSeq platform (Illumina, Inc., San Diego, CA, USA), according to the original Earth Microbiome Project protocols [31]. The index barcodes and adapter sequences were then removed from the pair-ended demultiplexed reads for downstream analysis.

### 2.4. Probiotic Mixture

All the participants received a daily capsule of a novel probiotic E3 formula developed by BioMed Microbiome Research Centre (BioMed Laboratory Company Limited, Hong Kong) containing a mixture of 8 types of highly effective gastro-resistant probiotics strains (no less than 2 × 10^11^ CFU/capsule at the time of production), prebiotics including inulin and oligosaccharide powder, and a postbiotic derived from *Lactobacillus plantarum*. This product was not designed as a single strain but as a probiotic mixture product with *Bifidobacterium* and *Lactobacilli*, as previous studies have shown that the effects of multi-strain probiotics are more effective than single-strain probiotics [32]. The probiotics mixture was composed of *Lactobacillus plantarum* GKM3, *Lactobacillus brevis* GKL9, *Lactobacillus acidophilus* GKA7, *Bifidobacterium bifidum* GKB2, *Lactobacillus casei* GKC1, *Lactobacillus reuteri* GKR1, *Bifidobacterium longum* GKL7, and *Lactobacillus gasseri* GKG1. *B. bifidum*, *B. longum*, *L. acidophilus*, and *L. plantarum* were shown to alleviate the pain and severity of patients with irritable bowel syndrome (IBS) in previous studies [33].

### 2.5. Bioinformatics Analysis

Only participants with (1) gut microbiome data and (2) Bristol Stool Form Scale (BSFS), (3) Psoriasis Area and Severity Index (PASI), and (4) Dermatology Life Quality Index (DLQI) scores available before and after the 8-week probiotic intervention had their information retrieved and were included in this analysis. The microbiome bioinformatics data were analyzed using the QIIME2-2023.5 [34], integrating various microbiome analysis algorithms and tools. Analysis of Compositions of Microbiome with Bias Correction (ANCOM-BC2) was used to perform the differential abundance analysis using a linear regression model [35].

### 2.6. Statistical Analysis

All statistical analysis and result visualization were conducted in R 4.3.1 and Python 3.10.12 and the *p*-value in this study was calculated using the R package ggpubr 0.6.0. Normality assumptions were evaluated using D’Agostino and Pearson’s tests. The Shapiro–Wilk test was employed for parametric tests. The demographic characteristics were evaluated using the non-parametric Mann–Whitney U rank test for continuous variables and Fisher’s exact test for categorical variables. The paired *t*-test or paired Wilcoxon signed-rank test was used to determine the differences between paired variables at week 0 and week 8. *p*-value correction was calculated using the Benjamini–Hochberg method. A *p*-value smaller than 0.05 was considered statistically significant unless otherwise specified.

## 3. Results

### 3.1. Study Cohort

A total of 58 participants between the ages of 18 and 65 years old diagnosed with psoriasis were initially recruited for this study. However, 13 participants were later excluded from the analysis due to withdrawal or incomplete recording of the BSFS scores. Thus, 45 participants aged between 25 and 63 with psoriasis and psoriasis-associated constipation were included in the analysis. The baseline demographic and disease characteristics are summarized in Table 1. The diagnosis and severity of psoriasis were evaluated by a board-certified dermatologist (S.K.F.L.). The participants were allocated into responder and non-responder groups based on their BSFS score at week 8. Those with BSFS scores maintained within or reaching the range of 3–5 at week 8 were regarded to be responders, while the others were allocated to the non-responder group. There were no statistically significant differences in sex, age, BMI, and psoriasis severity distributions between the responders and non-responders (sex: *p* = 0.0721; age: *p* = 0.9840; BMI: *p* = 0.0672; psoriasis severity: *p* = 0.7363). However, the prevalence of functional constipation and diarrhea between the responders and non-responders was significantly different at baseline (*p* = 0.0004). Furthermore, the responder group demonstrated a significantly lower impact of psoriasis on their quality of life, as indicated by the DLQI score (*p* = 0.0366).

### 3.2. Clinical Efficacy of Probiotic Intervention in GI-Related Comorbidities in Psoriasis Patients

Following an 8-week probiotic intervention, a higher proportion of participants had their BSFS score fall into the normal range when compared to the baseline at week 0. Unsurprisingly, the BSFS score did not exhibit a unidirectional change after the intervention (*p* = 0.3953, Figure 1A), as the BSFS is not a continuous scale of measurement. However, there was a significant difference in the distribution of the “Normal” and “Constipation/Diarrhea” groups between week 0 and week 8 based on the BSFS score (*p* = 0.0467, Figure 1B). A total of 10 out of 21 psoriasis patients (47.6%, 95% CI: 28.3% to 67.6%) with an abnormal BSFS at week 0 showed improvement by week 8, while a total of 34 out of 45 psoriasis patients (75.6%, 95% CI: 61.2% to 85.9%) achieved a normal BSFS score by week 8.

### 3.3. Gut Microbial Profile of the Psoriasis Patients

No significant difference in the gut microbial profile between week 0 and week 8 was observed, regardless of the participants’ response. At the phylum level, the top five most abundant phyla were *Firmicutes*, *Bacteroidota*, *Actinobacteriota*, *Proteobacteria*, and *Fusobacteriota* (Figure 2A). However, there was no significant change in the phylum level or Firmicutes/Bacteroidota (F/B) ratio after the 8-week course of probiotics (*p* = 0.5018, Figure 2B). No significant difference in the alpha diversity regardless of response was observed after the 8-week probiotic intervention (Appendix A).

Significant differences in the gut microbial profile between the responders and non-responders at week 0 and week 8 were observed. A total of 3628 unique amplicon sequence variants (ASVs) were identified at week 0, of which 104 are rare ASVs with only one count in any given sample included in this analysis. These ASVs belonged to 12 phyla, 20 classes, 49 orders, 90 families, 241 genera, and 570 species. The participants in the responder group tended to harbor more ASVs than the non-responder group (Figure 2C). Compared with the non-responders, the gut microbiome of the responders tended to harbor higher alpha diversity. (ACE Index: *p* = 0.1223; Chao 1 Index: *p* = 0.1288; Faith’s PD: *p* = 0.1823; observed OTUs: *p* = 0.1191; Shannon Diversity: *p* = 0.1500; Simpson Diversity: *p* = 0.1043; Appendix A). After the 8-week probiotic intervention, significantly higher diversities were observed in the responders in terms of the ACE Index (*p* = 0.0256), Chao 1 Index (*p* = 0.0274), observed OTUs (*p* = 0.0274), and Shannon Diversity (*p* = 0.0208) (Figure 3A). The ADONIS test also reported a statistically significant difference in beta diversity between responders and non-responders based on the Hamming distance (*p* < 0.001), Bray–Curtis (*p* < 0.001) dissimilarity, Jaccard index (*p* < 0.001), cosine similarity (*p* < 0.001), and unweighted UniFrac distance (*p* = 0.002) (Figure 3B and Appendix A).

### 3.4. Differential Abundance of ASVs after the 8-Week Probiotic Intervention

ANCOM-BC2 at the ASV level was employed to interrogate the impact of the probiotics on the gut microbiome profile. Seven ASVs were identified as correlating solely with timepoint (Figure 4A and Table 2). Three ASVs (DA-ASV1: *Lactobacillus acidophilus*, adjusted *p* = 0.0023; DA-ASV2: *Parabacteroides distasonis*, adjusted *p* < 0.001; and DA-ASV4: Ruminococcaceae CAG-352, adjusted *p* < 0.001, BLAST-aligned to *Ruminococcus bromii*) were enriched in the samples at week 8, while four ASVs (DA-ASV3: *Oscillibacter*, adjusted *p* = 0.0031; DA-ASV5: *Bacteroides vulgatus*, adjusted *p* < 0.001; DA-ASV6: *Escherichia* sp., adjusted *p* = 0.0273; and DA-ASV7: *Bilophila wadsworthia*, adjusted *p* = 0.0242) were depleted after the 8-week probiotic intervention. The differential abundance of these seven ASVs was further confirmed using the Wilcoxon signed-rank test on the centered log ratio (clr)-transformed abundance in the paired samples with *p* < 0.001 (Figure 5).

In addition, one ASV (DA-ASV8: *Fusicatenibacter*, adjusted *p* = 0.0147, BLAST aligned to *Fusicatenibacter saccharivorans*, Figure 6) was identified to be enriched in the responder group at week 0 and week 8 (Figure 4B and Appendix A). Although there was no significant difference between the centered log ratio (clr) transformed abundance in either responder or non-responder groups after the probiotics supplementation (Figure 7C,D), the differences between the responders and non-responders at either timepoint were statistically significant (Figure 7A,B). The extent of the difference between the non-responders and responders at week 8 was also larger than that at week 0 (mean fold change: week 0 = 1.730, 95% CI: 0.967 to 4.894; week 8 = 2.854, 95% CI: 1.355 to 33.556). The functions and impacts of these ASVs of differential abundance on human health may provide insights into the improvement of GI comorbidities and the varying responses to the 8-week probiotic intervention, as detailed in the discussion section.

## 4. Discussion

Despite the beneficial effects of the current psoriasis management plan on skin condition, gut homeostasis and psoriasis-related GI comorbidities remain out of reach using the conventional treatments. In this study, we focus on the potential clinical effects of probiotics on the GI comorbidities in psoriasis patients based on their symptomatic improvement, a gut microbiome evolution after an 8-week probiotic intervention, and the gut microbiome signatures in the responder group.

Firstly, a significant improvement in the GI comorbidities after the 8-week probiotic intervention was observed in the psoriasis patients according to their BSFS score. The proportion of normal GI conditions was significantly higher after the probiotic intervention, suggesting the potential benefits of probiotics in the improvement of psoriasis-related GI comorbidities. Unsurprisingly, there was no unidirectional change in the BSFS score after the probiotic intervention, as the BSFS is not measured on a continuous scale. Nevertheless, the unequal sample sizes of the responders and non-responders may lead to a loss of statistical power. Further clinical study with more balanced sample sizes is needed to validate the beneficial effects of probiotics on GI comorbidities.

The gut microbiome plays a crucial role in maintaining immune homeostasis in various immune-mediated diseases, including psoriasis [36]. Reductions in the gut microbiome’s diversity and richness are also found to be correlated with various human diseases, such as allergies, Crohn’s disease, ulcerative colitis, and colorectal disease [37]. Our study revealed a significantly different gut microbiome composition between the responders and non-responders after 8 weeks of probiotic intervention, along with improved GI comorbidities. We observed a significantly higher alpha diversity in the responders after the 8-week probiotic intervention when compared to the non-responders. In addition, the beta diversity between the responders and non-responders before and after the probiotic supplementation was also significantly different. These findings suggest probiotic intervention can potentially shift the gut microbiome composition to a higher diversity and richness. Therefore, this highlights the positive effects of probiotics in restoring gut microbiome homeostasis, which can potentially mediate an improvement in psoriasis-related GI comorbidities. However, the difference in the gut microbiome evolution between psoriasis patients and healthy people after probiotic intervention is unclear; further clinical study including non-psoriasis participants is warranted to validate the gut-homeostasis-promoting effects of probiotics.

We identified several differentially abundant ASVs before and after the 8-week course of probiotic intervention that may potentially contribute to the observed symptom improvements. We revealed that *Lactobacillus acidophilus*, *Parabacteroides distasonis*, and *Ruminococcus bromii* were significantly enriched after the intervention, while the relative abundance of *Oscillibacter*, *Bacteroides vulgatus*, *Escherichia* sp., and *Bilophila wadsworthia* was depleted after the 8-week probiotic intervention.

Previous studies have highlighted the health-promoting functions of *L. acidophilus*, including its role in improving obesity, controlling diabetes mellitus, the prevention of GI comorbidities, and the modulation of immune response [38,39,40]. It can suppress the overgrowth of pro-inflammatory bacteria, such as *Salmonella enteritidis* and *Staphylococcus aureus*, and the levels of toxic and pro-inflammatory metabolites secreted by these bacteria in the gut can therefore be reduced, and a healthy gut microenvironment can be restored [41,42]. The anti-inflammatory effect of *L. acidophilus* may also have positive effects on skin health by inhibiting the entry of pro-inflammatory metabolites into the skin cells via the “gut–skin axis”, thereby reducing skin inflammation and alleviating psoriasis. Similarly, *P. distasonis* also exerts protective effects in colitis patients through its membrane components, which decrease the levels of pro-inflammatory cytokines in the gut and reduce the severity of gut inflammation, therefore improving GI comorbidities [43]. Another study also observed a relatively low abundance of *P. distasonis* in psoriasis patients when compared to healthy controls [44]. A low abundance of *P. distasonis* may lead to the downregulation of the Th17 immune response [45].

On the other hand, the downregulation of pathogenic microorganisms might help to restore the balance of the gut microbiome. For *B. vulgatus*, a pathobiont associated with GI comorbidities like inflammatory bowel disease (IBD), its abundance was found to be relatively high in patients with ulcerative colitis [46]. The proteases and elastases secreted by *B. vulgatus* may increase the permeability of the intestinal membrane, leading to the invasion of pathogenic materials into the gut microenvironment, which can bring about harmful effects on gut health and lead to the progression of GI comorbidities [47,48]. *B. wadsworthia*, which is a sulfidogenic bacteria, has a prevalence rate found to be higher in the diseased population [49]. It can trigger a systemic inflammatory response by metabolizing sulfated compounds and producing hydrogen sulfide, which can directly exert toxic effects on the gut barrier, compromising the integrity of the gut membrane, increasing the risks of invasion by harmful bacteria and metabolites, and eventually leading to GI comorbidities [50]. By reducing the abundance of these harmful bacteria using probiotic intervention, the gut health of psoriasis patients can be improved, and the symptoms of gastrointestinal comorbidities and psoriasis can potentially be mitigated. In addition, we also identified enriched levels of *Fusicatenibacter saccharivorans* in the responders when compared to the non-responders. This observation was supported by previous studies, in which *F. saccharivorans* was found to possess anti-inflammatory properties, and its level is also negatively correlated with GI comorbidities like ulcerative colitis [51,52].

It is important to note that the probiotic formula used in this study not only contains probiotics but also prebiotics, including galacto-oligosaccharides (GOS), fructo-oligosaccharides (FOS), and inulin. Previous studies found that these prebiotics can potentially stimulate the growth of *Lactobacillus* and *Bifidobacterium* species [53]. The nourishing effects of prebiotics on probiotics can therefore reinforce the effectiveness of the probiotics to improve the symptoms of GI comorbidities and psoriasis. Furthermore, the fermentation of prebiotics can also stimulate the production of short-chain fatty acids (SCFAs) in the gut, such as butyrate, propionate, and acetate, which can suppress intestinal inflammation and regulate gut motility to improve constipation [54]. Thus, the inclusion of prebiotics in the probiotic formula may have also potentially magnified the effects of the probiotic intervention in the improvement of GI comorbidities and psoriasis.

In summary, this novel probiotic formula with prebiotics and postbiotics was noted to exert positive effects on functional GI comorbidities and the quality of life in patients with psoriasis, which may be neglected in the current psoriasis management plan. We have also identified several differentially abundant gut microbial signatures, which may help to explain the observed improvements in gut and skin health. These findings provide further support for the potential beneficial therapeutic effects of oral probiotic supplementation in managing psoriasis patients with associated functional GI comorbidities. Of note, our current study was limited by its small sample size and the absence of a placebo group. A further randomized placebo-controlled clinical study using an intention-to-treat (ITT) approach is warranted to validate the clinical implications of probiotics in the management of psoriasis in greater detail.

## Figures and Tables

**Figure 1 microorganisms-12-00208-f001:**
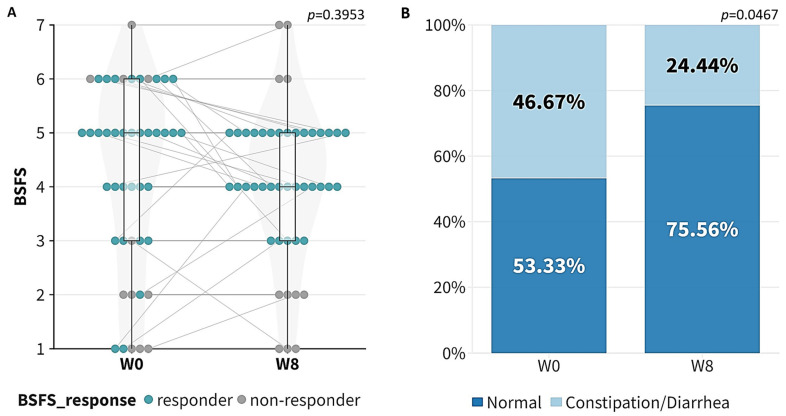
Improvement in Bristol Stool Form Scale (BSFS) after taking probiotics for 8 weeks. (**A**) Boxplot of BSFS score at week 0 (W0) and week 8 (W8); (**B**) proportion of normal and constipation/diarrhea patients at week 0 and week 8. Grey line indicated the paired data points at week 0 (W0) and week 8 (W8).

**Figure 2 microorganisms-12-00208-f002:**
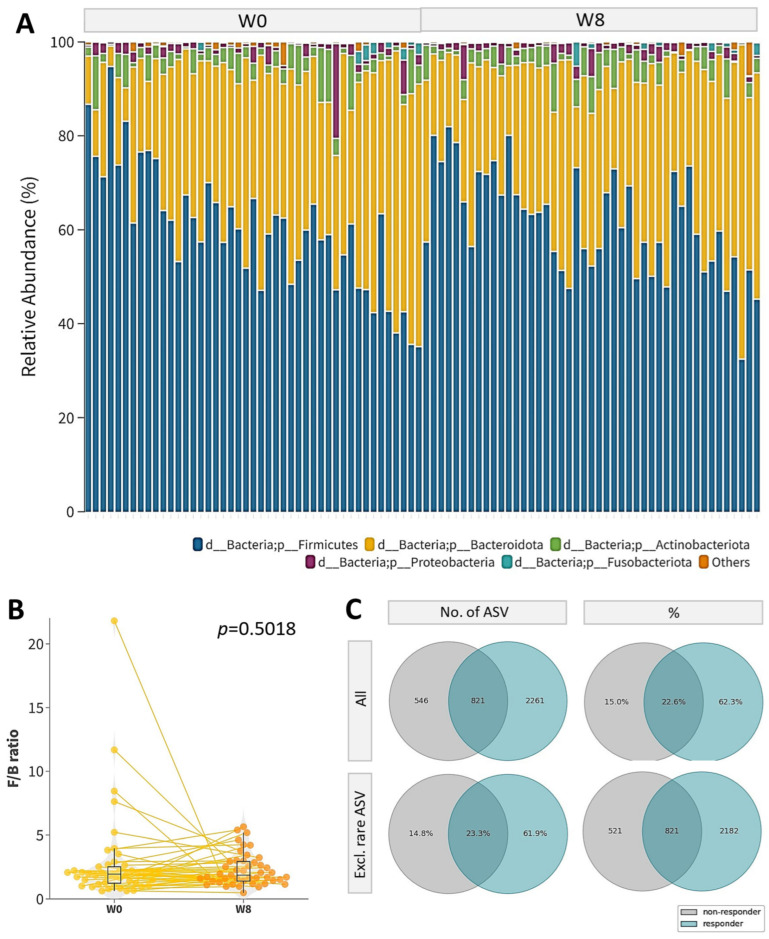
Gut microbiome composition of psoriasis patients. (**A**) Relative abundance of phyla at week 0 (W0) and week 8 (W8); (**B**) boxplot of Firmicutes/Bacteroides (F/B) ratio at week 0 and week 8; Yellow line indicated the paired data points at week 0 (W0) and week 8 (W8); (**C**) Venn diagram of number (**left**) and percentage (**right**) of all ASVs (**top**) and with rare ASVs excluded (**bottom**) within responder and non-responder groups at week 0. Rare ASVs were defined as ASVs that occurred in only one sample.

**Figure 3 microorganisms-12-00208-f003:**
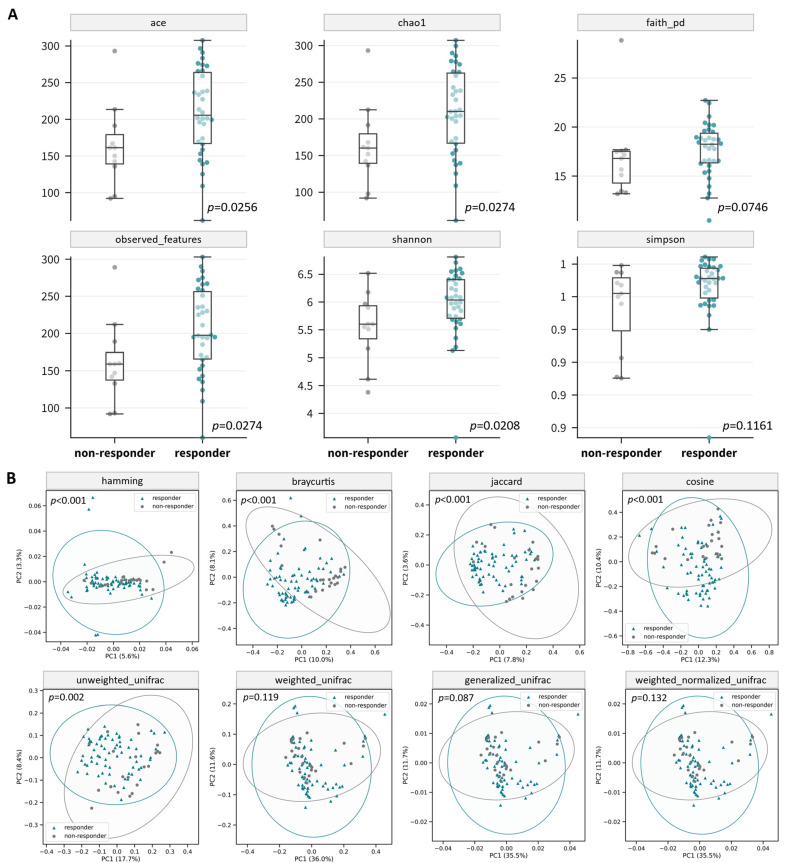
Gut microbiome composition of responder and non-responder groups of psoriasis patients. (**A**) Boxplots of alpha diversity at week 8; (**B**) principal coordinate analysis (PCoA) plots of beta diversity at week 0 and at week 8.

**Figure 4 microorganisms-12-00208-f004:**
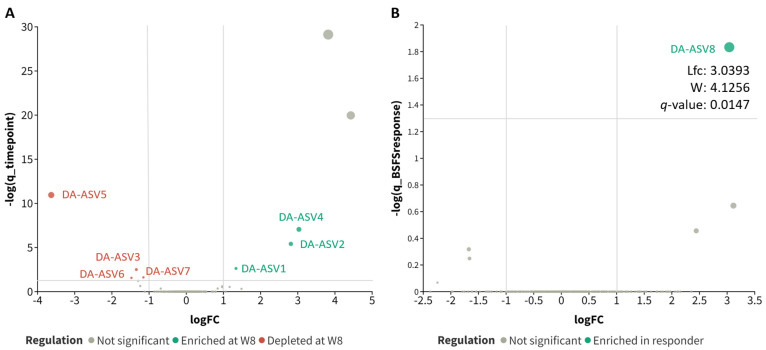
Differentially abundant ASVs using ANCOM-BC2 with significant *q* values. (**A**) Differentially abundant ASVs correlated solely with timepoint; (**B**) differentially abundant ASVs correlated solely with probiotics response. Log fold change (Lfc), test statistics (W). The taxon names of the DA-ASV are listed in Appendix A. The size of dots was proportional to the −log q values.

**Figure 5 microorganisms-12-00208-f005:**
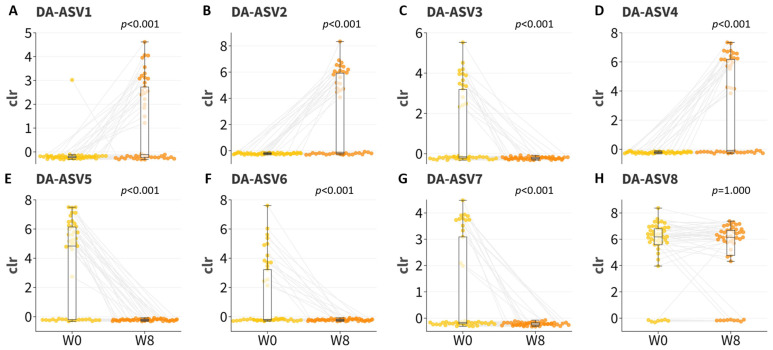
Boxplots of centered log ratio (clr) transformed abundance of differentially abundant ASVs correlated solely with timepoint at week 0 (W0) and at week 8 (W8). (**A**) DA-ASV1; (**B**) DA-ASV2; (**C**) DA-ASV3; (**D**) DA-ASV4; (**E**) DA-ASV5; (**F**) DA-ASV6; (**G**) DA-ASV7; (**H**) DA-ASV8. Grey line indicated the paired data points at week 0 (W0) and week 8 (W8).

**Figure 6 microorganisms-12-00208-f006:**
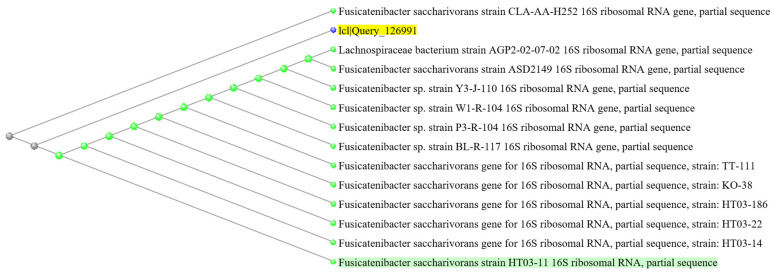
Phylogenetic tree of DA-ASV8 computed using MOLE-BLAST by using the 16S rRNA sequence as the search set with default parameter settings. Yellow highlighted text indicated the query and green highlighted text indicated the closest neighbors. The blue, grey and green dot indicated the query, untyped and typed strains, respectively.

**Figure 7 microorganisms-12-00208-f007:**
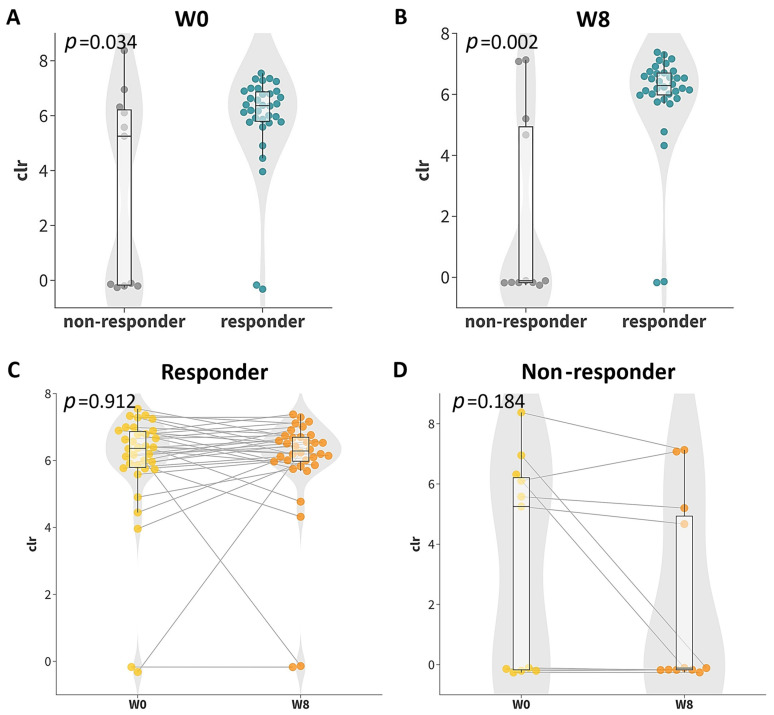
Boxplots of centered log ratio (clr)-transformed abundance of differentially abundant ASVs correlated solely with probiotics response (DA-ASV8: ab094218743ac1a1e8e57fad028130fd, *Fusicatenibacter saccharivorans*): (**A**) boxplot of responders and non-responders at week 0 (W0); (**B**) boxplot of responders and non-responders at week 8 (W8); (**C**) boxplot of responders at week 0 and week 8; (**D**) boxplot of non-responders at week 0 and week 8. Grey line indicated the paired data points at week 0 (W0) and week 8 (W8).

**Table 1 microorganisms-12-00208-t001:** Baseline demographic and disease characteristics of patients.

Variable	Responder (*n* = 34 ^)	Non-Responder (*n* = 11)	*p*-Value
Sex, no. (%)			
Male	24 (70.6)	4 (36.4)	0.0721
Female	10 (29.4)	7 (63.6)	
Age, mean (SD), y	44.5 (11.8)	44.8 (10.5)	0.9840
BMI, mean (SD) †	26.4 (5.1)	23.4 (4.4)	0.0672
Severity, no. (%) &			
Mild	15 (44.1)	4 (36.4)	0.7363
Moderate to severe	19 (55.9)	7 (63.6)	
BSFS, mean (SD)	4.5 (1.4)	3.4 (2.4)	0.2150
Constipation, no. (%)	3 (8.8)	6 (54.5)	0.0004
Normal, no. (%)	23 (67.6)	1 (9.1)	
Diarrhea, no. (%)	8 (23.5)	4 (36.4)	
BSA, mean (SD)	10.9 (9.9)	17.7 (27.0)	0.4681
PASI, mean (SD)	6.4 (4.5)	9.2 (10.0)	0.1092
DLQI, mean (SD)	10.4 (6.2)	15.9 (7.4)	0.0366
Psoriatic arthritis	11 (32.4)	2 (18.2)	0.4666

BMI, body mass index; PASI, Psoriasis Area and Severity Index; DLQI: Dermatology Life Quality Index. † BMI between 25.0 and 29.9 kg/m^2^ is classified as overweight, while BMI > 29.9 kg/m^2^ is classified as obese. & = someone either on biologics or with a PASI > 20 at baseline was regarded as having moderate to severe psoriasis. ^ = someone who had maintained or reached a BSFS in the range of 3–5 at week 8.

**Table 2 microorganisms-12-00208-t002:** Differentially abundant ASVs (taxon assigned using q2-feature-classifier) identified using ANCOM-BC2 after the 8-week course of orally administrated probiotics.

Feature ID	Taxon	Lfc	W	*q*-Value
3e75da00f0b8fefa3c1370c8b0600a95	g: *Lactobacillus*|s: *Lactobacillus acidophilus*	1.3380	4.5273	0.0023
721a741bef4a5db0211de1a5b84a8b5b	g: *Parabacteroides*|s: *Parabacteroides distasonis*	2.8137	5.7336	<0.001
85e43edd6eb1507d74c8b81f6448865f	f: *Oscillospiraceae*|g: *Oscillibacter*	−1.3427	−4.4644	0.0031
8cec479da287209d3c7d464f14242794	g: CAG-352|s: uncultured bacterium	3.0310	6.3449	<0.001
bd4606ad663e209e745c8c51b4deeee8	g: *Bacteroides*|s: *Bacteroides vulgatus*	−3.6316	−7.6051	<0.001
c3103408bd64a50afc4af497658f5164	g: *Escherichia*-Shigella|s: *Escherichia* sp.	−1.4750	−3.9759	0.0273
fe0d9776e545350d61d2a2bc6ebbc559	g: *Bilophila*|s: *Bilophila wadsworthia*	−1.1528	−4.0051	0.0242

## Data Availability

The raw sequence data are available from the NCBI (PRJNA934420). Due to the restriction of consent and sensitivity, the metadata are available upon reasonable request being made to the corresponding authors.

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
