# Peer review of "A Novel Multi-Strain E3 Probiotic Formula Improved the Gastrointestinal Symptoms and Quality of Life in Chinese Psoriasis Patients"

_microorganisms, 2024, doi:10.3390/microorganisms12010208_

Round 1
Reviewer 1 Report
Comments and Suggestions for Authors
The study is interesting, and the illustrations are bright. The methods for evaluation of stools, psoriatic signs, and the symptoms that deteriorate the QoL are adequate.
The blinding of assessors for skin lesions and of the stools was not achieved.
The methods in stool bacteria evaluation seems to be adequate. But the results are not very much impressive.
The abstract seemed to be adequate, the same impression for the introduction, the methods, the results.
The Discussion part is not very balanced, the part with the explanation of some bacteria roles is expanded in detriment of the results of the clinical trial (although some results were already published).
Formal analysis:
First, I think the number of psoriasis patients worldwide is wrong (2% of 8 billion would mean 160 million patients). -line 45
Line 58 -it is a non-logical mention of psoriasis, psoriasis - other than psoriasis.
Line 304 -L acidophilus is anti-inflammatory by TNFalpha (I thought that it is a proinflammatory cytokine).
Line 298 is contradictory. After the intervention (administration of the pro-post-pre biotic)” Oscillibacter, Bacteroides vulgatus, Escherichia sp., and Bilophila wadsworthia increased in occurrence in microbioma of the gut of the patients with psoriasis” - …possible deleterious effects?
Content analysis:
Limitations of the study – in my opinion (should be stated very clearly in the Discussion item):
1) The retrospective allocation in two arms: responders and not responders to the intervention; the size of the responder and non-responder groups was strikingly different (with different proportions of patients with constipation in non-responder patients, the division of patients in non-responders and responders being made also by the constipation improvement).
2) The analysis is not in ITT.
3) The lack of a comparative group treated with placebo.
4) The natural evolution of the treatment with the given pro, pre and postbiotic in non -psoriatic patient is also unknown.
5) The number of patients in non-responder group in small, the power of the test for detection of differences was too small (was it calculated?)
The ”inflation” of alpha risk was taking into account?
Then, the comparison between groups of non-responders and responders gives an unexpected significancy at the parameter constipation.
The Discussion part, maybe I must repeat, was not very much addressed to the findings of the clinical trial.
This fragment should be reformulated (lines 277-282):
”Patients with milder psoriasis tended to show better improvement in functional GI comorbidities after an 8-week probiotics intervention, although this correlation was not statistically significant. Furthermore, the BSFS score also significantly improved after intervention irrespective of the severity of psoriasis. These findings suggest a potential positive effect of probiotics in improving psoriasis-related GI comorbidities”.
I suggest mentioning the facts that are statistically significant only, and a phrase like the next one should be used instead of the former:
” Further studies are needed to assess if the (milder) psoriasis cutaneous manifestations and GI comorbidities could be improved with an 8-th week probiotics intervention…. The present study could lead to hypotheses that have to be verified in a larger scale randomized double blind trial.”

Reviewer 2 Report
Comments and Suggestions for Authors
The manuscript is interesting but it needs improvements.
Title
The title is lengthy, it could be shortened.
Abstract
Abstract did not have substantial results in the form of digits. I would the authors to modify and follow this structure in the abstract; background and problem, the rationale for the study; research objectives; some methodology; important data including statistical analysis; conclusions, novelty, and the importance of the findings.
Introduction
The paper fails to introduce novel concepts, ideas, or findings in its respective scientific field. The content appears to reiterate existing knowledge without presenting any substantial advancements or innovations. This lack of unannounced novelty significantly diminishes the paper's potential contribution to the scientific community.
Objectives of this study are not clear as background and research gaps are not identified
Background of the study is missing that why the developent Probiotic Formula to Improved Gastrointestinal Symptoms required how is different from other treatments or drugs?
Materials and methods
Ln 133-141: Please justify why probiotic mixture was used with the targeted concentration.
Which strains were used?
Results and Discussion section
please improve the resolution of the figures.
Discussion section is weak and could be elaborated with logical reasonings and further latest citations.
All figures are needed to be completed by complete details of abbreviations.
The figures quality must improve.
Conclusion section is weak. The conclusion section could be further summarized providing the major findings.
Please check the spaces between numbers and units all along the manuscript.
